# *TMEM132C* rs7296262 Single-Nucleotide Polymorphism Is Significantly Associated with Nausea Induced by Opioids Administered for Cancer Pain and Postoperative Pain

**DOI:** 10.3390/ijms25168845

**Published:** 2024-08-14

**Authors:** Yuna Kang, Daisuke Nishizawa, Seii Ohka, Takeshi Terui, Kunihiko Ishitani, Ryozo Morino, Miyuki Yokota, Junko Hasegawa, Kyoko Nakayama, Yuko Ebata, Kyotaro Koshika, Tatsuya Ichinohe, Kazutaka Ikeda

**Affiliations:** 1Addictive Substance Project, Tokyo Metropolitan Institute of Medical Science, Setagaya-ku, Tokyo 156-8506, Japan; kang-yn@igakuken.or.jp (Y.K.); nishizawa-ds@igakuken.or.jp (D.N.); ohka-si@igakuken.or.jp (S.O.); hasegawa-jk@igakuken.or.jp (J.H.); nakayama-kk@igakuken.or.jp (K.N.); ebata-yk@igakuken.or.jp (Y.E.); 2Department of Dental Anesthesiology, Tokyo Dental College, Chiyoda-ku, Tokyo 101-0061, Japan; 3Department of Neuropsychopharmacology, National Institute of Mental Health, National Center of Neurology and Psychiatry, Tokyo 187-8551, Japan; koshikakyotarou@tdc.ac.jp (K.K.); ichinohe@tdc.ac.jp (T.I.); 4Division of Internal Medicine, Department of Medicine, Higashi-Sapporo Hospital, Sapporo 003-8585, Japan; terui@hsh.or.jp (T.T.); ishitani@hsh.or.jp (K.I.); 5Division of Anesthesiology, Koujinkai Daiichi Hospital, Tokyo 125-0041, Japan; ryozomorino@daiichi.or.jp; 6Department of Anesthesiology, Cancer Institute Hospital, Tokyo 135-8550, Japan; yokota@jfcr.or.jp; 7Department of Anesthesiology, East Hokkaido Hospital, Kushiro 085-0036, Japan

**Keywords:** *TMEM132C*, SNP, nausea, chronic opioid use, acute phase of opioid use

## Abstract

Opioids are almost mandatorily used for analgesia for cancer pain and postoperative pain. Opioid analgesics commonly induce nausea as a side effect. However, the genetic factors involved are still mostly unknown. To clarify the genetic background of individual differences in the occurrence of nausea during opioid administration, the incidence of nausea was investigated in 331 patients (Higashi-Sapporo Hospital [HS] group) who received morphine chronically for cancer pain treatment and in 2021 patients (Cancer Institute Hospital [CIH] group) who underwent elective surgery under general anesthesia. We conducted a genome-wide association study of nausea in HS samples. Among the top 20 candidate single-nucleotide polymorphisms (SNPs), we focused on the *TMEM132C* rs7296262 SNP, which has been reportedly associated with psychiatric disorders. The rs7296262 SNP was significantly associated with nausea in both the HS and CIH groups (TT+TC vs. CC; HS group, *p* = 0.0001; CIH group, *p* = 0.0064). The distribution of nausea-prone genotypes for the rs7296262 SNP was reversed between HS and CIH groups. These results suggest that the *TMEM132C* rs7296262 SNP is significantly associated with nausea during opioid use, and the effect of the SNP genotype on nausea is reversed between chronic and acute phases of opioid use.

## 1. Introduction

Postoperative nausea and vomiting (PONV) is generally seen after major surgery. An estimated 30% of surgical patients will suffer from PONV during the first postoperative day [1]. Opioids are also used to treat cancer pain. Although opioids are the main treatment for cancer pain, opioid-related side effects, such as nausea and vomiting, may interfere with pain management and impair the quality of life of cancer patients [2]. Previously identified risk factors for PONV in adults include female sex, a history of PONV and/or motion sickness, nonsmoking status, and young age [3]. Anesthesia-related risk factors for PONV include volatile anesthetics, nitrous oxide, and postoperative opioids [4]. However, even patients at low PONV risk may suffer PONV, suggesting a genetic predisposition [5].

Opioids stimulate the medullary chemoreceptor trigger zone (CTZ), enhance vestibular sensitivity, and affect the gastrointestinal tract [2]. Nausea and vomiting are caused by various stimuli that act on the “vomiting center” in the medulla oblongata of the brain [6]. Four major areas—the CTZ, gastrointestinal tract, vestibular apparatus in the temporal lobe, and cerebral cortex—project to the vomiting center [6]. Opioids exert emetic effects, mainly through three mechanisms (i.e., direct stimulation of the CTZ, the inhibition of intestinal motility, and stimulation of the vestibular apparatus) [7]. Neurokinin-1 (NK-1) receptor, 5-hydroxytryptamine-2A (5-HT_2A_) receptor, and 5-HT_3_ receptor are ubiquitously expressed in human gastrointestinal vagal afferents and brain areas that are related to the vomiting reflex, such as the nucleus of the solitary tract [8]. Substance P is an endogenous ligand of NK-1 receptor and triggers NK-1 receptor signaling, causing nausea and vomiting [9]. Other receptors that are involved in nausea include dopamine D_2_ and D_3_ receptors and μ-opioid receptor [1].

Various single-nucleotide polymorphisms (SNPs) have been associated with nausea from acute or chronic opioid use, including *OPRM1* rs9397685 [10], *CHRM3* rs2165870 [11], and *KCNB2* rs349358 [12] for acute opioid use and *HTR3B* rs1176744 [13], *COMT* rs165722 [13], *CHRM3* rs10802789 [13], and *HTR3B* rs1672717 [13] for chronic opioid use. However, no SNPs have yet been reported to be commonly associated with nausea among acute and chronic opioid use. To clarify the general genetic background of individual differences in the occurrence of nausea during opioid administration, we investigated SNPs and their characteristics that are common to nausea during acute and chronic opioid administration.

## 2. Results

### 2.1. Impact of Clinical Variables on the Incidence of Nausea in Patients Who Were Treated with Opioids during the Treatment of Cancer Pain and in Patients Who Underwent Elective Surgery under General Anesthesia (Higashi-Sapporo Hospital [HS] and Cancer Institute Hospital [CIH] Groups)

In this study, samples with chronic opioid administration for cancer pain (HS group, *n* = 331) and samples with acute opioid administration in the perioperative period (CIH group, *n* = 2021) were analyzed. The background characteristics of the patients related to the surgery and anesthesia in the HS and CIH groups are shown in Appendix A.

In the HS group, 331 patients were evaluated for nausea, 42% of whom developed nausea (with nausea: 138, without nausea: 193). In the CIH group, all 2021 patients were evaluated for nausea, 42% of whom developed nausea (with nausea: 850, without nausea: 1171). Next, we investigated the relationship between patient characteristics and the presence or absence of nausea using a Mann–Whitney-*U* test or *χ*^2^ test. The following items showed significant associations with the incidence of nausea: in the HS group, age (*p* = 1.918 × 10^−2^), morphine [mg] (*p* = 4.650 × 10^−4^), morphine (normalized with body weight) [mg/kg] (*p* = 1.448 × 10^−4^; Table 1); in the CIH group, gender (*p* = 5.640 × 10^−17^), height (*p* = 4.815 × 10^−9^), weight (*p* = 1.223 × 10^−7^), body mass index (*p* = 2.758 × 10^−3^), smoking history (*p* = 9.555 × 10^−9^), motion sickness (*p* = 4.604 × 10^−11^), total dosage of fentanyl (*p* = 1.217 × 10^−9^), total dosage of remifentanil (*p* = 1.250 × 10^−5^), use of pentazocine (*p* = 5.688 × 10^−18^), use of opioid after anesthesia (*p* = 3.817 × 10^−15^), anesthesia method (*p* = 2.605 × 10^−7^), and pain (*p* = 1.340 × 10^−2^; Table 2).

Logistic regression analysis of the presence or absence of nausea was performed for items that were significantly associated with the phenotype among the patient characteristics, and odds ratios (ORs) were calculated. The following items were significantly associated with nausea: in the HS group, age (*p* = 3.176 × 10^−2^, Table 3); in the CIH group, gender (*p* = 5.640 × 10^−17^), smoking history (*p* = 9.555 × 10^−9^), motion sickness (*p* = 4.604 × 10^−11^), total dosage of fentanyl (*p* = 1.074 × 10^−7^), total dosage of remifentanil (*p* = 1.583 × 10^−4^), use of pentazocine (*p* = 5.688 × 10^−18^), use of opioid after anesthesia (*p* = 3.817 × 10^−15^), anesthesia method (*p* = 2.605 × 10^−7^), and pain (*p* = 1.340 × 10^−2^; Table 4). In the CIH group, the following items had large ORs and were strongly associated with nausea: gender (OR = 2.281, 95% confidence interval [CI] = 1.877–2.773), use of pentazocine (OR = 2.252, 95% CI = 1.870–2.712), and use of opioid after anesthesia (OR = 2.045, 95% CI = 1.709–2.447; Table 4).

### 2.2. Exploration of Genetic Polymorphisms Associated with Nausea during Treatment of Cancer Pain by Genome-Wide Association Study (GWAS) in HS Samples

We comprehensively explored genetic variants that were associated with the presence or absence of nausea in the total of 331 patient subjects in the HS group who were treated with opioids for cancer pain. A total of 648,817 SNPs that met the quality control standards in the GWAS were examined for relationships with the phenotypes in the trend, dominant, and recessive models for minor alleles for each SNP. However, no SNPs showed genome-wide significant associations with the presence or absence of nausea, with the lowest *p* = 3.369 × 10^−6^ for the rs7282115 SNP in the trend model (Table 5). The top 20 candidate SNPs that were selected from the GWAS for nausea in the HS group did not include the reported SNPs that were associated with nausea (i.e., *OPRM1* rs9397685 [10], *CHRM3* rs2165870 [11], *KCNB2* rs349358 [12], *HTR3B* rs1176744 [13], *COMT* rs165722 [13], *CHRM3* rs10802789 [13], and *HTR3B* rs1672717 [13]).

### 2.3. Association between TMEM132C rs7296262 SNP and Nausea in Patients Who Were Treated with Opioids for Cancer Pain and in Patients Who Underwent Elective Surgery under General Anesthesia (HS and CIH Samples)

Among the top 20 gene-annotated SNPs in the HS GWAS with regard to the association with nausea (Table 5), we selected the *TMEM132C* rs7296262 SNP (i.e., the only SNP that was investigated in a previous study of a psychiatric disorder at the survey stage on 24 May 2024) [14]. The *TMEM132C* rs7296262 SNP is associated with bipolar disorder (BD) with a history of suicide attempts [14,15]. Physical symptoms are prevalent in BD (47.8%) [16] and may be an independent risk factor of disease severity and suicidal ideation [17]. Among physical symptoms, gastrointestinal symptoms, including nausea, have a strong relationship with the anxiety that BD patients experience during a depressive episode [18]. Thus, the *TMEM132C* rs7296262 SNP may feasibly be associated with nausea. To investigate this possibility, we performed statistical association analyses of the *TMEM132C* rs7296262 SNP using SPSS 28 software in the HS and CIH groups.

The genotypic distribution of the *TMEM132C* rs7296262 SNP did not significantly deviate from the theoretical Hardy–Weinberg equilibrium (HWE; HS group: *p* > 0.05, *χ*^2^ = 1.339; CIH group: *p* > 0.05, *χ*^2^ = 3.396). Linkage disequilibrium (LD) analysis was performed for HS and CIH samples (Appendix A, respectively). The rs7296262 SNP was present in an intron region. In the HS group, there were four other SNPs within the LD block that includes the rs7296262 SNP, all of which showed strong LD (*D*’ = 1) with each other (12:129095258, 12:129095945, kgp5245701, rs73161919). There were two SNPs with strong LD other than the LD block that includes the rs7296262 SNP (rs12321675, 12:129092485). All of these SNPs are located in intron regions. In the CIH group, there were two other SNPs within the LD block that includes the rs7296262 SNP, all of which showed strong LD with each other (rs10744383, rs1466642). There was one SNP with strong LD other than the LD block that includes the rs7296262 SNP (rs2220486). All of these SNPs are located in intron regions.

The *TMEM132C* rs7296262 SNP showed a significant association with nausea in the genotypic model in the HS and CIH groups (HS group: *p* = 7.000 × 10^−4^, Table 6; CIH group: *p* = 2.010 × 10^−2^, Table 7). In the recessive model, the *TMEM132C* rs7296262 SNP was significantly associated with nausea (HS group: *p* = 1.000 × 10^−4^, Table 6; CIH group: *p* = 6.400 × 10^−3^, Table 7). No significant association was found in the dominant model (HS group: *p* > 0.05, Table 6; CIH group: *p* > 0.05, Table 7). The prevalence of nausea among genotypes was then analyzed. In the HS group, a higher rate of nausea was observed in CC carriers than in TT+TC carriers (TT+TC/CC; with nausea, 77%/23%; without nausea, 92%/8%; Table 6). In the CIH group, a higher rate of nausea was observed in T-allele carriers (TT+TC/CC: with nausea, 88%/12%; without nausea, 83%/17%; Table 7). The results suggest that the nausea-prone genotype of the rs7296262 SNP was reversed in the HS and CIH groups.

To further investigate the effect of opioid use on the incidence of nausea, differences in the genotype distribution of opioid use were analyzed. For the amount of opioid use, the amount of opioids equivalent to oral morphine in the HS group and the amount of fentanyl use in the CIH group were analyzed. No significant difference was found in the genotype distribution of opioid use among genotypes in the HS and CIH groups (HS group: genotypic model, dominant model, recessive model, *p* > 0.05, Table 8; CIH group: genotypic model, dominant model, recessive model, *p* > 0.05, Table 9).

## 3. Discussion

We conducted a GWAS of nausea in the HS group (Table 5). Among the top 20 gene-annotated SNPs in the results of the HS GWAS in the recessive model with regard to the association with nausea, we selected the *TMEM132C* rs7296262 SNP for further analysis, which is reportedly associated with psychiatric disorders. The *TMEM132C* rs7296262 SNP was significantly associated with nausea during opioid use. In the HS group that chronically received opioids for cancer pain, the rs7296262 SNP was significantly associated with nausea in the recessive model. A higher rate of nausea was observed in CC carriers than in TT+TC carriers (Table 6). In the CIH group that received acute opioid administration during general anesthesia, the rs7296262 SNP was significantly associated with nausea in the recessive model. A higher rate of nausea was observed in TT+TC carriers than in CC carriers (Table 7). Thus, the nausea-prone genotype of the rs7296262 SNP was reversed with acute and chronic opioid use. These results suggest that the *TMEM132C* rs7296262 SNP is involved in the mechanisms of nausea during opioid use and the genotype reversal phenomenon between acute and chronic opioid use.

The *TMEM132C* rs7296262 SNP was significantly associated with nausea induced by opioids, consistent with the *TMEM132C* rs7296262 association with BD with suicide attempts, which are associated with nausea [14,15,16,17,18]. This implies that the *TMEM132C* rs7296262 SNP may also contribute to nausea in BD patients with suicide attempts.

Postoperative opioid use has been reported to increase the risk of PONV [19]. μ-Opioid receptor agonists, the most potent analgesics [20], were used for intraoperative anesthesia, and the κ-opioid receptor agonist pentazocine (30 mg) was administered for postoperative pain in the CIH group in this study [21]. Pentazocine was reported to cause a 17.2% incidence of nausea in a group that received 30 mg pentazocine [22], which is a lower rate than that caused by opioids in the present study (42% in both the HS and CIH groups), implying a minimal contribution of pentazocine to nausea. In the present study, there was a significant difference between the use and absence of use of pentazocine in the CIH group, depending on the presence or absence of nausea (Table 2), whereas the genotype distribution of the *TMEM132C* rs7296262 SNP showed no significant differences between the use and absence of use of pentazocine in the CIH group (Table 9). This means the same pentazocine administration rate induces nausea at different rates, depending on the genotype of the *TMEM132C* rs7296262 SNP. Although the contribution of pentazocine to nausea is minimal, we cannot exclude the possibility that the rs7296262 genotype is involved in nausea induced by the κ-opioid receptor agonist pentazocine.

The search for genomic functional regions revealed that human permissive enhancers exist around 6 kilobase pairs (kbp) upstream of the *TMEM132C* rs7296262 SNP (Appendix A) [15], and the cap-analysis gene expression (CAGE) signal that is associated with active promoters and enhancers exists around 6 kbp upstream of the *TMEM132C* rs7296262 SNP (Appendix A) [15]. Additionally, protein-coding transcripts have been reported that start from 67 kbp upstream and 83 kbp downstream of the rs7296262 SNP [23] (Appendix A). Thus, the *TMEM132C* rs7296262 SNP may be associated with transcriptional activity for *TMEM132C* protein-coding transcripts. However, the *TMEM132C* rs7296262 SNP may also be associated with full-length transcripts or other transcripts. Further research is required to clarify the precise function of the *TMEM132C* rs7296262 SNP.

The *TMEM132C* rs7296262 SNP, which was associated with nausea in the present study, has been reported to be involved in BD and suicide [24]. One of the essential receptors related to nausea and vomiting is the NK-1 receptor. NK-1 receptor expression significantly decreased in monocytes in BD patients compared with healthy subjects [25]. Furthermore, 12 depressed patients, including 6 who committed suicide, showed lower NK-1 receptor expression in the orbitofrontal cortex compared with controls in a postmortem study [26]. Altogether, the *TMEM132C* rs7296262 SNP could be associated with BD and suicide, as well as nausea through NK-1 receptor signaling, although further research is needed.

NK-1 and 5-HT_2A_ receptors are related to opioid-induced nausea and vomiting [1]. Acute morphine treatment was reported to upregulate the functional expression of NK-1 receptor in cortical neurons in morphine-treated rats [27], whereas chronic morphine treatment decreased substance P and NK-1 receptor immunoreactivity in the dorsal horn in morphine-treated rats [28]. Moreover, chronic morphine exposure increased 5-HT_2A_ receptor [29]. Thus, acute or chronic morphine treatment influences the expression of NK-1 and 5-HT_2A_ receptors and possibly has opposite effects on NK-1 receptor expression. In the present study, the distribution of nausea-prone genotypes for the *TMEM132C* rs7296262 SNP was reversed between the CIH and HS samples (i.e., between acute and chronic opioid administration), implying an association with the effect of acute or chronic morphine on these nausea-related receptors. However, further research is required to elucidate the mechanisms by which the *TMEM132C* rs7296262 SNP influences opioid-induced nausea through NK-1 and 5-HT_2A_ receptor signaling.

Acute opioid administration is less likely to cause nausea in homozygote carriers of the C allele of the rs7296262 SNP, whereas chronic opioid administration is more likely to cause nausea in homozygote carriers of the C allele of the rs7296262 SNP, based on the present study. The allele frequencies of the rs7296262 SNP of the *TMEM132C* gene in different regional populations in the present study were the following: in the HS group, T-allele frequency of 64% and C-allele frequency of 36%; in the CIH group, T-allele frequency of 62% and C-allele frequency of 38%. The frequencies in the HS and CIH groups are similar to East Asian populations (T-allele frequency of 61% and C-allele frequency of 39%) and other regional populations (e.g., American populations: 57% T allele, 43% C allele; European populations: 55% T allele, 45% C allele; South Asian populations: 65% T allele, 35% C allele), except for African populations according to the 1000 Genomes study in the SNP database [24]. African populations show a T-allele frequency of 44% and C-allele frequency of 56%, resulting in small frequency reversals for African populations and others. These results suggest that the allele frequencies that are associated with nausea and vomiting with acute and chronic opioid administration have no large variations among regional populations.

The present study has limitations. First, some patients received the κ-opioid receptor agonist pentazocine in addition to opioids that mainly have affinity for μ-opioid receptor. Thus, the present study could not distinguish between the contributions of the different opioid receptor subtypes. Second, some patients received the volatile anesthetic desflurane for the maintenance of general anesthesia. The use of volatile anesthetics is associated with a risk of nausea [4]. Other supplementary analgesics, including other painkillers that have affinity for serotonin, norepinephrine, and dopamine receptors, were administered at the discretion of primary care doctors if required. Thus, the supplementary analgesics and their dosages were variable among patients. The present study cannot eliminate that possible unexpected side effects of these supplementary analgesics led to nausea.

## 4. Materials and Methods

### 4.1. Patients

#### 4.1.1. Patients Who Were Treated with Opioids during the Treatment of Cancer Pain in the HS Group

We enrolled 428 adult Japanese patients (20–94 years old, 213 males and 215 females) who suffered from various types of cancer and were hospitalized at Higashi-Sapporo Hospital (Hokkaido, Japan) for the treatment of cancer pain in 2017–2019. Because the presence or absence of nausea could not be evaluated in 97 of the 428 patients, the analysis was performed for 331 patients (20–94 years old, 163 males and 168 females). Higashi-Sapporo Hospital specializes in cancer care, particularly palliative care [30,31]. All of the patients who were recruited in the present study were treated with opioid analgesics, and many were also appropriately treated with nonsteroidal anti-inflammatory drugs (NSAIDs) and/or other supplementary analgesics for the treatment of pain. We excluded patients who were considered unsuitable by their primary care doctors. Detailed demographic and clinical data of the subjects were provided in a previous report [32]. Peripheral blood samples were collected from these subjects for the gene analysis.

The study was conducted according to guidelines of the Declaration of Helsinki and approved by the Institutional Review Board or Ethics Committee of Higashi-Sapporo Hospital and Tokyo Metropolitan Institute of Medical Science (Tokyo, Japan; protocol code: 17-1). Written informed consent was obtained from all patients.

#### 4.1.2. Patients Who Underwent Elective Surgery under General Anesthesia in the CIH Group

Enrolled in the study were 2021 adult patients (20–94 years old, 700 males and 1321 females) who were scheduled to undergo elective surgery for cancer under general anesthesia by TIVA with propofol or inhalational anesthesia with desflurane at The Cancer Institute Hospital of the Japanese Foundation for Cancer Research (CIH samples). The 2021 patients consisted of the previously reported 806 patients [21] and an additional 1215 patients. As detailed in the previous report [21], the exclusion criteria were the following: (1) patients to whom mild or more emetogenic antitumor agents were administered or who were scheduled to be administered from 6 days before the start of the study to 48 h after surgery; (2) patients with symptomatic brain metastases; (3) patients who used the following antiemetic drugs within 48 h before and during surgery: 5-HT_3_ receptor antagonists (granisetron, ondansetron, azasetron, etc.), phenothiazines (chlorpromazine, prochlorperazine, perphenazine, etc.), butyrophenone-based preparations (haloperidol, droperidol, etc.), benzamide preparations (sulpiride, tiapride, sultopride, etc.), dopamine receptor antagonists (metoclopramide, itopride, domperidone, etc.), antihistamines (hydroxyzine, dimenhydrinate, diphenhydramine), or NK-1 receptor antagonists (apireptant); (4) patients who were mentally unable to communicate; (5) patients who were pregnant; (6) patients who were judged to be inappropriate for inclusion in the study by the investigator; and (7) patients of the Head and Neck Department and Gastroenterology Department who needed advanced management in the postoperative intensive care unit. The major reasons for applying these exclusion criteria were the possible influence of these factors on the incidence and severity of PONV and the collection of accurate data. The cancellation criteria were the following: (1) patients for whom blood collection was not possible and (2) patients whose informed consent was withdrawn. All of the individuals who were included in the study were of Japanese origin. Peripheral blood samples were collected from these subjects for gene analysis. Detailed demographic and clinical data of the subjects were provided in a previous report [21]. The study was conducted according to the guidelines of the Declaration of Helsinki and was approved by the Institutional Review Board or Ethics Committee of The Cancer Institute Hospital and Tokyo Metropolitan Institute of Medical Science (Tokyo, Japan; protocol code: 21-17). Written informed consent was obtained from all patients.

#### 4.1.3. Patient Characteristics and Clinical Data for the HS Group

We obtained data on surgical history, treatment history, pain status (e.g., presence/absence of somatic pain, visceral pain, and neuropathic pain), drug treatments, and disease status (e.g., lung cancer, breast cancer, stomach cancer, etc. [32]). Some of the patients were afflicted with multiple diseases.

The treatment of pain was mainly conducted by administering opioid analgesics, including morphine, oxycodone, fentanyl, tapentadol, tramadol, methadone, and hydromorphone, which mainly activate the μ-opioid receptor [32]. Various types of drugs, such as NSAIDs (e.g., loxoprofen and diclofenac) and/or other supplementary analgesics (e.g., pregabalin and dexamethasone), were also administered at the discretion of primary care doctors if required. To allow intersubject comparisons of the opioid analgesic doses that were required for cancer pain treatment, the opioid doses were converted to equivalent doses of oral morphine, as described in a previous report [32]. The total dose of converted opioid analgesics administered was calculated daily, and the total dose of analgesics was calculated as a daily average based on the amount of 5 days of administration, 3–7 days before blood collection. This average total dose was used as the endpoint of opioid requirements for the genetic association analysis in the present study. Doses of analgesics that were administered were normalized to body weight. The detailed clinical data of the subjects are presented in Appendix A.

#### 4.1.4. Patient Characteristics and Clinical Data for the CIH Group

We obtained data on patient characteristics (gender, age, height, weight, and body mass index), history of smoking, history of motion sickness, clinical data for the postoperative period (type of anesthesia, total dose of remifentanil, total dose of fentanyl, postoperative administration of opioid drugs, and postoperative opioid administration [including pentazocine]), the experience and frequency of postoperative pain, and nausea (Appendix A). The μ-opioid receptor agonists fentanyl and remifentanil were administered intraoperatively, and the κ-opioid receptor agonist pentazocine was administered for postoperative analgesia [21].

### 4.2. Whole-Genome Genotyping and Quality Control

For the HS and CIH samples, a total of 428 and 806 DNA samples from the patients, respectively, were used for genotyping. Total genomic DNA was extracted from whole-blood samples using standard procedures. The extracted DNA was dissolved in TE buffer (10 mM Tris-HCl and 1 mM ethylenediaminetetraacetic acid, pH 8.0). The DNA concentration was adjusted to 50 ng/μL and 100 ng/μL for whole-genome genotyping for the HS and CIH samples, respectively, using a NanoDrop ND-1000 Spectrophotometer (NanoDrop Technologies, Wilmington, DE, USA).

According to the manufacturer’s recommendations, whole-genome genotyping was performed by utilizing the Infinium Assay II with an iScan system (Illumina, San Diego, CA, USA), as described in previous reports [21,32]. An Infinium Asian Screening Array-24 v. 1.0 BeadChip was used to genotype all HS samples (total markers: 659,184). Three versions of HumanOmniExpressExome-8 Bead-Chips (v. 1.2, total markers: 964,193; v. 1.3, total markers: 958,497; v. 1.4, total markers: 960,919) were used to genotype 806 of the CIH samples. Approximately 946,000 common SNP markers were included in the three BeadChips versions. Numerous copy number variation markers were included in the BeadChips, but the majority of the Bead-Chips were for SNP markers on the human autosome or sex chromosome.

GenomeStudio with the Genotyping v. 2.0.4 module (Illumina) was used to examine data for samples that had their entire genomes genotyped to assess the quality of the findings. Following data cleaning, samples with genotype call rates less than 0.95 were not included in the remaining studies. No patient samples were consequently discarded for subsequent investigations (i.e., all of the samples met the 0.95 quality control cutoff, and all samples were considered for further association analyses). In the subsequent association analyses, markers with genotype call frequencies less than 0.95 and “Cluster sep” (a measure of genotype cluster separation) values less than 0.1 were not included. Markers were further filtered based on a test of HWE. Markers with *p* values (*df* = 1) less than 0.001 were considered to be deviated in the HWE tests and thus were excluded. After these filtration procedures, the patient samples retained a total of 648,817 and 651,086 SNP markers, as described in previous reports [21,32].

For the HS samples, all whole-genome genotyping data that passed the quality control criteria were used for the GWAS. For the CIH samples, only the genotype data for the selected rs7296262 SNP in the *TMEM132C* gene region among all of the genotyped SNPs were extracted and used for further association analyses.

### 4.3. TaqMan Genotyping

After whole-genome genotyping, we performed an additional TaqMan assay on 1215 CIH samples. The TaqMan allelic discrimination assay was conducted for genotyping the rs7296262 SNP, as described in previous reports [33,34]. To perform the TaqMan assay with a LightCycler 480 II (Roche Diagnostics, Basel, Switzerland), we used TaqMan SNP Genotyping Assays (Life Technologies, Carlsbad, CA, USA) that contained sequence-specific forward and reverse primers to amplify the polymorphic sequence and two probes that were labeled with VIC and FAM dye to detect both alleles of the rs7296262 SNPs (Assay ID: C_1179719_10). Real-time polymerase chain reaction was performed in a final volume of 10 μL that contained 2× LightCycler 480 II Probes Master (Roche Diagnostics), 40× TaqMan SNP Genotyping Assays, 5–50 ng genomic DNA as the template, and H_2_O (Roche Diagnostics). The thermal conditions were the following: 95 °C for 10 min, followed by 45 cycles of 95 °C for 10 s and 60 °C for 60 s, with final cooling at 50 °C for 30 s. Afterward, endpoint fluorescence was measured for each sample well, and each genotype was determined based on the presence or absence of each type of fluorescence.

### 4.4. Statistical Analysis

For the HS group, the presence or absence of nausea during the treatment period with opioids was evaluated. For the CIH group, the presence or absence of nausea during the 48 h postoperative period was evaluated. In the association studies, these criteria were used as indices of the vulnerability to nausea.

To explore associations between SNPs and the incidence of nausea, Fisher’s exact test or the Cochran–Armitage trend test was conducted for the HS samples, and genotype data were compared between subjects with and without the incidence of nausea. Trend, dominant, and recessive genetic models were used for the analyses. Male genotypes were not included in the analysis of X chromosome markers, whereas both male and female individuals were included in the association study for autosomal markers. PLINK v. 1.07 (https://zzz.bwh.harvard.edu/plink/index.shtml; accessed 30 October 2023) [35], gPLINK v. 2.050 [36], and Haploview v. 4.1 [37] were used to perform the statistical analyses and visualize the results. As in the previous GWAS of opioid analgesic requirements in the HS samples [32], the criterion for significance in the GWAS was set to *p* < 5 × 10^−8^, which is widely known to be a conventional criterion for the level of significance in GWASs [37,38].

To validate associations between the rs7296262 SNP and the incidence of nausea in the HS samples, an *χ*^2^ test was conducted for CIH samples with SPSS 28 software (IBM Japan, Tokyo, Japan), and genotype data between subjects with and without the incidence of nausea were compared. Genotypic, dominant, and recessive genetic models were used for the analyses. The criterion for significance in the association analysis was set to *p* < 0.05. Additionally, HWE was tested using the *χ*^2^ test (*df* = 1) for genotypic distributions of the rs7296262 SNP, with values of significant deviation set to *p* = 0.05.

To estimate the impact of clinical and genetic factors on the incidence of nausea, we used a multivariate analysis (i.e., logistic regression analysis). Variables that were significantly associated with PONV were subsequently included in the multivariable logistic regression model using a stepwise forward selection strategy. The dependent variable in both the HS and CIH groups was nausea (Yes/No). The independent variables in the HS group were patient age and morphine oral equivalent amount. Independent variables in the CIH group were gender (male/female), height, weight, body mass index, smoking history (Yes/No), motion sickness (Yes/No), total dosage of fentanyl, total dosage of remifentanil, use of pentazocine (Yes/No), use of opioid after anesthesia (Yes/No), anesthesia method (TIVA/inhalation anesthetic), and pain (Yes/No).

### 4.5. Additional In Silico Analysis

#### 4.5.1. Power Analysis

Statistical power analyses were preliminarily performed using G*Power 3.1.3 software [39]. Power analyses for Fisher’s exact tests, with the allocation ratio set to 0.5, indicated that the expected power (1 minus type II error probability) was 80.0% for the type I error probability, which was set to 1.000 × 10^−7^ (closest to 5 × 10^−8^ in this software) when risk allele frequencies for patients with and without nausea were 0.3860 and 0.1000, respectively, and when the sample sizes for patients with and without nausea were 307 and 153, in the present study. However, for the same type I error probability and sample sizes of 310 and 155, the expected power decreased to 50.0% when the risk allele frequencies for patients with and without nausea were 0.3465 and 0.1000, respectively. Conversely, the estimated risk allele frequencies for patients with and without nausea were 0.4060 and 0.1000, respectively, for the same type I error probability, and sample sizes of 305 and 152 were required to achieve 90% power. Therefore, a single analysis in the present study might be expected to detect true associations with the phenotypes, with 80% statistical power for effect sizes from large to moderately medium but not small, although the exact effect size is poorly understood in cases of SNPs that significantly contribute to nausea.

#### 4.5.2. LD Analysis

Data from whole-genome-genotyped samples were extracted using GenomeStudio 2.0 with the Genotyping v. 3.3.7 module to assess quality of the results for SNPs within the TMEM132C gene region. There were no SNPs with low typing rates, and SNPs with a minor allele frequency (MAF) greater than 0 were extracted. LD analysis was performed on 168 SNPs in the TMEM132C gene region in the SNP array in the HS group and on 178 SNPs in the TMEM132C gene region in the SNP array in the CIH group. To estimate the LD intensity between SNPs, the commonly used D’ and r^2^ values were calculated pairwise using the genotype dataset for each SNP. The LD block was defined among SNPs that showed “strong LD” based on the default algorithm of Gabriel et al. [40] with an upper limit of 0.98 and a lower limit of 0.7 for the 95% CI of D’ that indicated strong LD. TagSNPs in the LD blocks were determined using the Tagger software package that is incorporated in Haploview 4.2, which was detailed in a previous report [41].

#### 4.5.3. Reference of Databases

Several databases and bioinformatic tools were referenced to more thoroughly examine the candidate SNP, which may be related to human opioid analgesic sensitivity, including the National Center for Biotechnology Information database on 31 January 2023 [24] and HaploReg v. 4.1 on 6 June 2024 [42,43]. HaploReg is a tool for investigating non-coding genomic annotations at variations in haplotype blocks, such as potential regulatory SNPs at disease-associated sites [42]. Data on human permissive enhancers and CAGE derived from FANTOM5 were extracted using ZENBU to investigate transcriptional regulation around the rs7296262 SNP [15] on 1 June 2024. Data on the transcripts starting around the rs7296262 SNP were extracted using European Molecular Biology Laboratory-European Bioinformatics Institute [23] on 6 June 2024.

## 5. Conclusions

The *TMEM132C* rs7296262 SNP was significantly associated with nausea during opioid use. The distribution of nausea-prone genotypes for the *TMEM132C* rs7296262 SNP was reversed between CIH and HS samples (i.e., between acute and chronic opioid administration), implying an association with the effect of acute or chronic morphine on these nausea-related receptors. However, further research is required to elucidate the mechanisms by which the *TMEM132C* rs7296262 SNP influences opioid-induced nausea signaling through nausea-related receptors.

## Figures and Tables

**Table 1 ijms-25-08845-t001:** Background characteristics of the patients and information related to the opioid administration and cancer pain of the HS group.

	With Nausea (*n* = 138)	Without Nausea (*n* = 193)	*p*
Gender (male/female)	66/72 (48%/52%)	97/96 (50%/50%)	6.624 × 10^−1^
Age (year)	69.413 ± 12.615	72.508 ± 11.934	1.918 × 10^−2^ *
Height (cm)	158.007 ± 8.828	157.558 ± 8.193	5.275 × 10^−1^
Weight (kg)	50.019 ± 11.745	51.109 ± 10.902	2.468 × 10^−1^
Drinking (+/−)	35/100 (26%/74%)	45/142 (24% / 76%)	7.028 × 10^−1^
Smoking history (+/−)	54/81 (40%/60%)	87/100 (47% / 53%)	2.443 × 10^−1^
Morphine (mg) ^a^	101.721 ± 161.967	71.625 ± 152.273	4.650 × 10^−4^ *
Morphine (mg/kg) ^b^	1.989 ± 2.863	1.464 ± 3.038	1.448 × 10^−4^ *
Neuropathic pain (+/−)	34/104 (25%/75%)	50/143 (26%/74%)	7.936 × 10^−1^

The data are expressed as numbers or the mean ± standard deviation (SD). * *p* < 0.05. ^a^ Total dose of analgesics as a daily average equivalent to oral morphine. ^b^ Total dose of analgesics as a daily average per body weight equivalent to oral morphine.

**Table 2 ijms-25-08845-t002:** Background characteristics of the patients and information related to the surgery and anesthesia of the CIH group.

	With Nausea (*n* = 850)	Without Nausea (*n* = 1171)	*p*
Gender (male/female)	206/644 (24%/76%)	494/677 (42%/58%)	5.640 × 10^−17^ *
Age (year)	57.035 ± 13.567	57.594 ± 13.836	2.747 × 10^−1^
Height (cm)	159.151 ± 7.664	161.339 ± 8.378	4.815 × 10^−9^ *
Weight (kg)	55.595 ± 11.310	58.011 ± 11.125	1.223 × 10^−7^ *
Body mass index (kg/m^2^)	21.882 ± 3.794	22.207 ± 3.447	2.758 × 10^−3^ *
Smoking history (+/−)	341/509 (40%/60%)	621/550 (53%/47%)	9.555 × 10^−9^ *
Motion sickness (+/−)	450/399 (53%/47%)	448/723 (38%/62%)	4.604 × 10^−11^ *
Total dosage of fentanyl (μg)	217.275 ± 131.081	184.952 ± 127.410	1.217 × 10^−9^ *
Total dosage of remifentanil (μg)	2889.765 ± 2391.726	2472.502 ± 2173.196	1.250 × 10^−5^ *
Use of pentazocine (+/−)	592/258 (70%/30%)	591/580 (50%/50%)	5.688 × 10^−18^ *
Use of opioid after anesthesia (+/−)	493/357 (58%/42%)	472/699 (40%/60%)	3.817 × 10^−15^ *
Anesthesia method (TIVA^a^/inhalation anesthetic)	218/632 (26%/74%)	427/744 (36%/64%)	2.605 × 10^−7^ *
Pain (+/−)	617/233 (73%/27%)	790/381 (67%/33%)	1.340 × 10^−2^ *

The data are expressed as numbers or the mean ± standard deviation (SD). * *p* < 0.05. ^a^ General anesthesia using total intravenous anesthesia (TIVA).

**Table 3 ijms-25-08845-t003:** Factors behind the incidence of nausea in the multivariate analysis in the HS group.

Variable	OR ^a^	95% CI ^b^	*p*
Age (year)	0.979	0.960–0.998	3.176 × 10^−2^ *
Morphine (mg) ^c^	1.002	0.996–1.008	5.693 × 10^−1^
Morphine (mg/kg) ^d^	0.946	0.683–1.308	7.351 × 10^−1^

* *p* < 0.05. ^a^ Odds ratio. ^b^ Confidence interval. ^c^ Total dose of analgesics as a daily average equivalent to oral morphine. ^d^ Total dose of analgesics as a daily average per body weight equivalent to oral morphine.

**Table 4 ijms-25-08845-t004:** Factors behind the incidence of nausea in the multivariate analysis in the CIH group.

Variable	OR ^a^	95% CI ^b^	*p*
Gender (male/female)	2.281	1.877–2.773	5.640 × 10^−17^ *
Height (cm)	1.037	0.962–1.117	3.460 × 10^−1^
Weight (kg)	1.010	0.912–1.119	8.504 × 10^−1^
Body mass index (kg/m^2^)	1.006	0.774–1.308	9.644 × 10^−1^
Smoking history (+/−)	0.593	0.496–0.712	9.555 × 10^−9^ *
Motion sickness (+/−)	1.820	1.522–2.177	4.604 × 10^−11^ *
Total dosage of fentanyl (μg)	0.998	0.997–0.999	1.074 × 10^−7^ *
Total dosage of remifentanil (μg)	1.000	0.998–0.999	1.583 × 10^−4^ *
Use of pentazocine (+/−)	2.252	1.870–2.712	5.688 × 10^−18^ *
Use of opioid after anesthesia (+/−)	2.045	1.709–2.447	3.817 × 10^−15^ *
Anesthesia method (TIVA ^c^/inhalation anesthetic)	1.664	1.370–2.021	2.605 × 10^−7^ *
Pain (+/−)	1.277	1.052–1.551	1.340 × 10^−2^ *

* *p* < 0.05. ^a^ Odds ratio. ^b^ Confidence interval. ^c^ General anesthesia using TIVA.

**Table 5 ijms-25-08845-t005:** Top 20 candidate SNPs selected from the GWAS for nausea in HS samples.

Model	Rank	CHR	SNP	Position	*p*	Related Gene	Genotype (Nausea +)				Genotype (Nausea −)		
							A/A	A/B	B/B		A/A	A/B	B/B

Trend	1	21	rs7282115	21159990	0.000003369	*NCAM2*	21	77	40		11	80	102
Trend	2	17	rs2671822	35270663	0.000008426	*-*	22	70	46		10	77	106
Trend	3	20	20:55479012	56903956	0.00001031	*-*	2	27	109		8	79	106
Trend	4	3	rs890914	159984019	0.00001213	*IL12A-AS1*	1	31	105		0	13	180
Trend	5	8	8:136946790	135934547	0.0000136	*-*	0	13	125		0	0	193
Trend	6	20	20:58319409	59744354	0.00001464	*PHACTR3*	4	47	87		1	30	159
Trend	7	4	JHU_4.133461451	132540297	0.00002577	*-*	18	69	51		61	90	42
Trend	8	2	2:142096588	141339019	0.00002735	*LRP1B*	9	53	76		2	46	145
Trend	8	2	JHU_2.142096799	141339231	0.00002735	*LRP1B*	9	53	76		2	46	145
Trend	10	18	18:38956090	41376126	0.00002754	*-*	2	28	107		12	72	108
Trend	11	11	rs10742466	39179258	0.00003006	*-*	0	12	126		0	0	193
Trend	12	12	rs1861371	97381156	0.00003233	*-*	7	46	85		26	91	76
Trend	13	2	rs2683834 *	141358516	0.00004015	*LRP1B*	9	51	77		2	45	146
Trend	14	2	kgp10749785	152514871	0.000042	*FMNL2*	2	35	101		1	17	175
Trend	15	7	rs650974 *	103756236	0.00004238	*RELN*	46	68	24		31	100	62
Trend	16	18	18:38953465	41373501	0.00004501	*-*	2	29	107		12	72	109
Trend	16	18	rs8089014	41374361	0.00004501	*-*	2	29	107		12	72	109
Trend	16	18	18:38954459	41374495	0.00004501	*-*	2	29	107		12	72	109
Trend	16	18	18:38955349	41375385	0.00004501	*-*	2	29	107		12	72	109
Trend	16	18	18:38958695	41378731	0.00004501	*-*	2	29	107		12	72	109
Trend	16	18	18:38960113	41380149	0.00004501	*-*	2	29	107		12	72	109
Trend	16	18	rs12454051	41380714	0.00004501	*-*	2	29	107		12	72	109


Dominant	1	20	20:55479012	56903956	0.000005041	*-*	2	27	109		8	79	106
Dominant	2	8	8:136946790	135934547	0.000008149	*-*	0	13	125		0	0	193
Dominant	3	21	rs7282115	21159990	0.00001736	*NCAM2*	21	77	40		11	80	102
Dominant	4	11	rs10742466	39179258	0.00002063	*-*	0	12	126		0	0	193
Dominant	4	11	11:39300012	39278462	0.00002063	*-*	1	11	126		0	0	193
Dominant	6	3	rs890914	159984019	0.00002983	*IL12A-AS1*	1	31	105		0	13	180
Dominant	7	20	20:58319409	59744354	0.00003028	*PHACTR3*	4	47	87		1	30	159
Dominant	8	10	rs3740337 *	86662394	0.00003319	*OPN4*	19	65	54		14	58	121
Dominant	8	10	rs17425484	86666593	0.00003319	*-*	19	65	54		15	57	121
Dominant	10	20	20:20330281	20349637	0.00003408	*CFAP61*	6	53	79		4	37	152
Dominant	11	10	rs7069923 *	18441439	0.00003631	*CACNB2*	45	74	19		40	88	65
Dominant	12	18	18:38956090	41376126	0.0000378	*-*	2	28	107		12	72	108
Dominant	13	2	kgp10749785	152514871	0.00003931	*FMNL2*	2	35	101		1	17	175
Dominant	14	2	kgp2534744 *	75111493	0.00004175	*TACR1*	1	34	103		1	15	177
Dominant	15	10	kgp1635663	86673416	0.00005721	*LDB3*	29	74	35		22	80	91
Dominant	16	21	rs2823824	16466449	0.00006737	*MIR99AHG*	0	2	136		0	25	168
Dominant	16	21	exm2272994 *	16505474	0.00006737	*MIR99AHG*	0	2	136		0	25	168
Dominant	18	18	18:38953465	41373501	0.00006826	*-*	2	29	107		12	72	109
Dominant	18	18	rs8089014	41374361	0.00006826	*-*	2	29	107		12	72	109
Dominant	18	18	18:38954459	41374495	0.00006826	*-*	2	29	107		12	72	109
Dominant	18	18	18:38955349	41375385	0.00006826	*-*	2	29	107		12	72	109
Dominant	18	18	18:38958695	41378731	0.00006826	*-*	2	29	107		12	72	109
Dominant	18	18	18:38960113	41380149	0.00006826	*-*	2	29	107		12	72	109
Dominant	18	18	rs12454051	41380714	0.00006826	*-*	2	29	107		12	72	109


Recessive	1	3	kgp9242183	107978309	0.00001306	*-*	17	51	70		2	66	125
Recessive	1	3	rs35887155	107986512	0.00001306	*-*	17	51	70		2	66	125
Recessive	3	23	JHU_X.9082224	9114184	0.00002276	*-*	32	24	16		14	52	30
Recessive	4	2	kgp8484512	47972189	0.00004133	*-*	15	79	44		57	80	55
Recessive	5	16	rs2075520 *	21211351	0.00004225	*ZP2*	35	51	52		16	104	72
Recessive	6	2	kgp8864616	47971375	0.00004236	*-*	15	80	43		57	80	56
Recessive	6	16	rs11075364	17561020	0.00004236	*-*	15	78	45		57	94	42
Recessive	8	23	rs6640292	9098151	0.00005369	*-*	32	24	16		15	51	30
Recessive	9	14	rs7147499	25334328	0.00007244	*-*	13	31	94		1	56	136
Recessive	10	2	rs17720710	147333809	0.000073	*-*	0	45	93		18	68	107
Recessive	11	4	JHU_4.133461451	132540297	0.00008035	*-*	18	69	51		61	90	42
Recessive	12	19	rs11083554	40360815	0.00008898	*PLD3*	12	70	56		49	83	61
Recessive	13	23	rs7876208	9083032	0.0001062	*-*	31	25	16		15	51	30
Recessive	14	2	2:172139165	171282655	0.0001304	*-*	10	40	88		0	58	135
Recessive	14	20	rs180479	5908546	0.0001304	*-*	10	31	97		0	51	142
Recessive	16	16	kgp16425294	77074647	0.0001489	*-*	0	49	89		16	54	123
Recessive	17	15	exm1158632 *	45153958	0.0001921	*DUOX1*	46	60	32		30	100	63
Recessive	18	19	19:35042712	34551807	0.0002114	*-*	2	59	77		23	78	92
Recessive	19	12	rs7296262 *	128610527	0.0002193	*TMEM132C*	32	54	52		16	89	88
Recessive	20	22	kgp2905464	29338808	0.0002205	*AP1B1*	25	51	62		10	73	110


Model, the genetic model in which candidate SNPs were selected by the GWAS; CHR, chromosome number; Position, chromosomal position (bp); Related gene, the nearest gene from the SNP site; A/A, homozygote for the minor allele for each SNP; A/B, heterozygote for each SNP; B/B, homozygote for the major allele for each SNP; * SNP for which genotype data were available in CIH samples.

**Table 6 ijms-25-08845-t006:** Effects of genetic models of *TMEM132C* rs7296262 SNP on nausea in the HS group.

Genetic Models and Nausea Occurrence		Genotypes		*p*
Genotypic model (TT, TC, CC)	TT [*n* = 140] (%)	TC [*n* = 143] (%)	CC [*n* = 48] (%)	
With nausea (*n* = 138)	52 (38)	54 (39)	32 (23)	7.000 × 10^−4^ *
Without nausea (*n* = 193)	88 (46)	89 (46)	16 (8)	

Dominant model (TT vs. TC+CC)	TT [*n* = 140] (%)	TC+CC [*n* = 191] (%)		
With nausea (*n* = 138)	52 (38)	86 (62)		1.507 × 10^−1^
Without nausea (*n* = 193)	88 (46)	105 (54)		

Recessive model (TT+TC vs. CC)	TT+TC [*n* = 283] (%)		CC [*n* = 48] (%)	
With nausea (*n* = 138)	106 (77)		32 (23)	1.000 × 10^−4^ *
Without nausea (*n* = 193)	177 (92)		16 (8)	

* *p* < 0.05.

**Table 7 ijms-25-08845-t007:** Effects of genetic models of *TMEM132C* rs7296262 SNP on nausea in the CIH group.

Genetic Models and Nausea Occurrence		Genotypes		*p*
Genotypic model (TT, TC, CC)	TT [*n* = 805] (%)	TC [*n* = 910] (%)	CC [*n* = 306] (%)	
With nausea (*n* = 850)	355 (43)	388 (45)	107 (12)	2.010 × 10^−2^ *
Without nausea (*n* = 1171)	450 (39)	522 (44)	199 (17)	

Dominant model (TT vs. TC+CC)	TT [*n* = 805] (%)	TC+CC [*n* = 1216] (%)		
With nausea (*n* = 850)	355 (42)	495 (58)		1.305 × 10^−1^
Without nausea (*n* = 1171)	450 (38)	721 (62)		

Recessive model (TT+TC vs. CC)	TT+TC [*n* = 1715] (%)		CC [*n* = 306] (%)	
With nausea (*n* = 850)	743 (88)		107 (12)	6.400 × 10^−3^ *
Without nausea (*n* = 1171)	972 (83)		199 (17)	

* *p* < 0.05.

**Table 8 ijms-25-08845-t008:** Amount of opioid required according to genetic models of *TMEM132C* rs7296262 SNP in the HS group.

Genetic Models and Amount of Opioid Required		Genotypes		*p*
Genotypic model (TT, TC, CC)	TT	TC	CC	
Morphine (mg/kg)	0.789 (0.419–1.861)	0.853 (0.441–1.463)	0.798 (0.382–1.492)	9.403 × 10^−1^

Dominant model (TT vs. TC+CC)	TT	TC+CC		
Morphine (mg/kg)	0.798 (0.419–1.861)	0.833 (0.417–1.463)		9.705 × 10^−1^

Recessive model (TT+TC vs. CC)	TT+TC		CC	
Morphine (mg/kg)	0.825 (0.428–1.623)		0.798 (0.382–1.492)	7.533 × 10^−1^

The data are expressed as the median (25th–75th percentiles).

**Table 9 ijms-25-08845-t009:** Amount of opioid required according to genetic models of *TMEM132C* rs7296262 SNP in the CIH group.

Genetic Models and Amount of Opioid Required		Genotypes		*p*
Genotypic model (TT, TC, CC)	TT	TC	CC	
Total dosage of fentanyl (μg)	200 (100–300)	200 (100–300)	200 (100–300)	9.303 × 10^−1^
Total dosage of remifentanil (μg)	2000 (1300–3500)	2000 (1200–3375)	2000 (1300–3400)	8.598 × 10^−1^
Use of pentazocine (+/−)	472/333	540/370	171/135	5.673 × 10^−1^
Use of opioid after anesthesia (+/−)	380/425	441/469	144/162	8.440 × 10^−1^

Dominant model (TT vs. TC+CC)	TT	TC+CC		
Total dosage of fentanyl (μg)	200 (100–300)	200 (100–300)		8.850 × 10^−1^
Total dosage of remifentanil (μg)	2000 (1300–3500)	2000 (1200–3400)		5.830 × 10^−1^
Use of pentazocine (+/−)	472/333	711/505		9.419 × 10^−1^
Use of opioid after anesthesia (+/−)	380/425	585/631		6.905 × 10^−1^

Recessive model (TT+TC vs. CC)	TT+TC		CC	
Total dosage of fentanyl (μg)	200 (100–300)		200 (100–300)	7.041 × 10^−1^
Total dosage of remifentanil (μg)	2000 (1200–3500)		2000 (1300–3400)	8.690 × 10^−1^
Use of pentazocine (+/−)	1012/703		171/135	3.065 × 10^−1^
Use of opioid after anesthesia (+/−)	821/894		144/162	7.931 × 10^−1^

The data are expressed as the median (25th–75th percentiles).

## Data Availability

The raw data in this study are shown in Appendix A.

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
