# Peer review of "TMEM132C rs7296262 Single-Nucleotide Polymorphism Is Significantly Associated with Nausea Induced by Opioids Administered for Cancer Pain and Postoperative Pain"

_ijms, 2024, doi:10.3390/ijms25168845_

Round 1

Reviewer 1 Report

Comments and Suggestions for Authors

The research investigates the genetic factors associated with nausea induced by opioid use for cancer pain and postoperative pain. It is argued that the TMEM132C rs7296262 SNP is significantly associated with nausea during opioid use, with different effects in chronic and acute phases. There are some issues to supplement.

1. Although the contents have been shown TMEM132C rs7296262 SNP is associated with nausea during opioid use. However, there was no straightforward evidence that TMEM132C rs7296262 SNP is directly involved in nausea. It is wondering whether the basis for including the TMEM132C rs7296262 SNP was only based on previous research that links to psychiatric disorders, bipolar disorder, and suicide. The author must provide a clear genetic or clinical basis for selecting the TMEM132C rs7296262 SNP among various SNP candidates in Table 5.

2. The results showed that the use of pentazocine and the use of opioids after anesthesia were significantly related to the onset of nausea. However, no significant relationship appears between the use of pentazocine and opioids after anesthesia and TMEM132C rs7296262 SNP. This may be due to the fact that it basically causes nausea by the use of pentazocine and the use of opioids after anesthesia and is not caused by genetic characteristics, such as TMEM132C rs7296262 SNP. The authors must discuss these points.

Comments on the Quality of English Language

Minor language polishing is required.

Author Response

添付ファイルをご覧ください。

Reviewer 2 Report

Comments and Suggestions for Authors

The current study conducted a genome-wide association study for nausea in patient samples and found that TMEM132C rs7296262 SNP is significantly associated with nausea during opioid use. Moreover, genotypes for the TMEM132C rs7296262 SNP were reversed between acute and chronic opioid ministration. See comments below.

Line 79 and Line 81: Either both use 42% or Forty-two percent.

Line 143- 145: Why were all 428 patients tested if only 331 patients in the HS group had nausea data (line 78)?

Line 157: There seem to be some sentences missing here.

The majority of the introduction needs to be paraphrased. For example, the first sentence was copied from https://journals.lww.com/md-journal/subjects/Anesthesiology/Fulltext/2015/05030/Neurokinin_1_Receptor_Antagonists_in_Preventing.4.aspx.

Reviewer 3 Report

Comments and Suggestions for Authors

Dear Authors

overall a very nice manuscript with very relevant findings. Only some minor comments could be done:

Comments to the introduction section: PONV is commonly seen after major surgeries where opioids are used for post surgery analgesia. In the clinical praxis it is also well known that nausea and vomiting strongly interfere with the used anaesthetics. Therefore, a direct comparison between the two pain states is not 100% correct, although the aesthetic risk factors are mentioned in this section. At least the interference with the medication cocktail in these status should be mentioned.

To use the term opioids in this discussion may also not precise enough to the involvement of different opioid receptors (mü, kappa, delta, ORL1) and the receptor functionality (e.g., partial antagonistic component of buprenorphine). However, morphine is the classical example of a mü-opioid.

The presentation of results is very well done. All relevant data are described and summarized in overview tables. The reader is drawn into the presentation of the data and all conclusions are precisely formulated. However, a comment and discussion on the treatment regime of morphine and used dosages should be added. The data in table 8 (morphine in mg/kg) alone are not sufficient. Another essential information is missing: In clinical praxis a mixture of analgesics is frequently used, Therefore, the information if other opioids beside morphine is given, is under represented in the discussion, although other opioids like fentanyl and cogeners ar listed. The post-surgery treatment regime could include other pain killers having affinity to serotonin, noradrenaline or dopamine receptors. These components may have an additional effect on the results. It is really recommended to include a short paragraph on these aspects, although in the “Materials and Methods” section pre- and post-surgery treatment options are discussed.

The discussion section is very well structured. Obtained data are evaluated and interpreted, the conclusions can be easily followed.

In the “Materials and Methods” section all relevant data and information’s are presented. Additional information are presented in the supplementary material file.

The statistical analysis is well performed and correctly presented. The conclusion are summarized very succinctly and the finding are of highly relevance to get more understanding and insight  in opioid side effects and a big step forward to personalized analgesia with less side effects in the acute and chronic settings.

Reviewer 4 Report

Comments and Suggestions for Authors

Thank you for permitting me to review this manuscript 

In this study the authors intended to further investigate SNP and their charaxteristics to better understand genetics of individual difference in occurrence of nausea during opioid administration 

they conclude TMEM132C rs7296262 SNP is highly associated  with nausea during opioid therapy 

I have some suggestion 

the authors should clarify , how these Snp related to psyhcotic disorders were selected and why did they think their hypothesis  turned to be true 

Table2 please display some percentage  to facilitate readers assimilation of the study 

table 5 was statistical comparison performed between nausea + and nausea - 

Please explain or elaborate this  paragraph in the discussion 

Acute opioid administration is less likely to cause nausea in homozygote carriers of 260 the C allele of the rs7296262 SNP, whereas chronic opioid administration is more likely to 261 cause nausea in homozygote carriers of the C allele of the rs7296262 SNP, based on the 262 present study.

Round 2

Reviewer 2 Report

Comments and Suggestions for Authors

Section 2.2 line 143-145: I understand 331 patients were evaluated for nausea for HS group. So how could you evaluate GWAS data association with nausea for all 428 patients, when only 331 patients have nausea data? Wouldn’t you only be able to that with the 331 patients, not all 428 patients (Line 144)?
